# The Establishment and Spread of a Newly Introduced Begomovirus in a Dry Tropical Environment Using Tomato Yellow Leaf Curl Virus as a Case Study

**DOI:** 10.3390/plants11060776

**Published:** 2022-03-14

**Authors:** Cherie Gambley, Peter Nimmo, Janet McDonald, Paul Campbell

**Affiliations:** Department of Agriculture and Fisheries, Brisbane 4001, Australia; peter.nimmo@daf.qld.gov.au (P.N.); janet.mcdonald@daf.qld.gov.au (J.M.); paul.campbell@daf.qld.gov.au (P.C.)

**Keywords:** begomovirus, TYLCV, whitefly, epidemiology, *Bemisia*

## Abstract

Early detection of tomato yellow leaf curl virus (TYLCV) in a previously unaffected tomato production district in Australia allowed its spread to be evaluated spatially and temporally. The population dynamics of the TYLCV vector, *Bemisia argentifolii* (silverleaf whitefly, SLW), were also evaluated. The district is a dry tropical environment with a clear break to commercial production during the summer wet season. The incidence of TYLCV within crops and its prevalence through the district was influenced by weather, location, vector movements, and the use of Ty-1 virus-resistant hybrids. Rainfall had an important influence, with late summer and early autumn rain suppressing the levels of SLW and, by contrast, a dry summer supporting faster population growth. The use of Ty-1 hybrids appears to have reduced the incidence of TYLCV in this district. There was limited use of Ty-1 hybrids during 2013, and by season end, crops had moderate levels of SLW and high virus incidence. The 2015 and early 2016 season had high SLW populations, but TYLCV incidence was lower than in 2013, possibly due to the widespread adoption of the Ty-1 hybrids reducing virus spread. This study provides valuable epidemiology data for future incursions of begomoviruses, and other viruses spread by SLW.

## 1. Introduction

The International Committee for the Taxonomy of Viruses (ICTV) currently recognises 408 species within the Begomovirus genus [1]. Begomoviruses are a concern for tropical and sub-tropical agricultural areas and an increasing problem in temperate protected cropping systems [2]. These viruses are transmitted in a persistent, circulative manner through the active or passive flight of infective (viruliferous) adult insects from whitefly species belonging to the *Bemisia* genus (*Hemiptera*: *Alyerodidae*) [3]. This includes silverleaf whitefly (SLW), which is widely distributed in Australian tropical and subtropical horticulture, cotton, and grains cropping regions [4], and as such, these major production areas are at risk of exotic begomovirus incursions. Begomoviruses threaten cotton, legume, solanaceous (chili, capsicum, eggplant, tomato, and potato), cucurbit (zucchini, pumpkin, melon, and squash), and sweet potato crops, with a multitude of viruses causing disease in many individual crops. In addition to *Bemisia argentifolii* (SLW; previously known as B-biotype and MEAM1-biotype of *B. tabaci*; [5]), biotypes of the *B. tabaci* species complex also vector begomoviruses, with varying transmission efficiencies, depending on the biotype-virus interaction. In Australia, the diversity of *Bemisia* spp. is thought to be very low and dominated by SLW [4,5,6]. However, whitefly populations within Australia are not routinely monitored.

Begomovirus diversity in Australia is extremely low, with only five known species. These are *tomato yellow leaf curl virus*–IL strain (TYLCV-IL) and *tomato leaf curl virus* (ToLCV), which largely only affect solanaceous crops [7]. ToLCV has very limited distribution within Australia, is not found outside northern Australia, and not in commercial crops [7]. The remaining three species are *sweet potato leaf curl virus* (SPLCV) which affects sweet potato crops, *Abutilon mosaic virus* (AbMV), and *honeysuckle yellow vein mosaic virus* (HYVMV) [7], and these species don’t infect tomato. SPLCV is not widespread in Australia and does not cause economic impacts. AbMV is used in the ornamental industry to produce variegated Chinese lantern plants and is no longer whitefly transmissible [8]. Similarly, HYVMV does not cause economic damage in Australia, although it is reported to cause tomato disease in Japan [9]. There are also no known detections of DNA-beta satellites in Australia [7].

Globally, in regions with a high diversity of begomoviruses, particularly where the viruses have common hosts, recombination between different individual virus species can occur and give rise to a new virus species with an altered host range. This makes control and management of the diseases a constant challenge, particularly if control is dependent on host resistance. The threat to Australia posed by begomovirus incursions is substantial, and its possible viruliferous whiteflies could be blown into northern Australia from other countries. Timor Leste and Papua New Guinea are known to have begomoviruses exotic to Australia [10]. The consensus on the long-distance movement of begomoviruses, including between countries or within countries, is due to the movement of infected plant material or viruliferous insect vectors [11,12,13,14,15,16,17,18,19].

TYLCV is highly destructive to tomato crops wherever it occurs [20]. TYLCV-Israel (TYCLV-IL) was first detected in Australia in 2006 [7] and was restricted to tomato production areas in the sub-tropical areas of southeast Queensland until 2011. At this time, TYLCV-IL moved to a dry tropical production area in northern Queensland, where it again caused significant economic impact [21]. The introduction of TYLCV into this new district is thought to be via infected seedlings from southeast Queensland. Locally produced seedlings are typically used in the dry tropics, however, in June 2010, 5–7 million seedlings were lost due to a poisoning incident at the local nursery (https://www.abc.net.au/news/2010-07-06/seedling-sabotage-devastates-farmers/894442; accessed on 24 January 2022). As a direct consequence, late-season plantings of 2010 seedlings were unavailable locally, and a large volume was sourced from districts where TYLCV was common. Diagnostics samples sent from the district in early 2011 confirmed the presence of TYLCV-IL.

Although host resistance, through the *Ty-1* resistance gene, helps reduce economic losses from TYLCV, there are few other known resistance genes to other begomovirus species, thus it is unlikely that host resistance will be available if new begomoviruses are introduced into Australia. Consequently, it was important to understand how begomoviruses establish and spread to mitigate losses from potential future incursions. The early detection of TYLCV in a commercial production district allowed investigation of the establishment and spread of this virus in a newly affected area. Furthermore, unlike the southeast, the dry tropical environment has a clear break in cropping due to the annual wet season, and the district is limited in alternative weed and crop hosts of the virus over this break [21]. These factors simplified analyses of spatial and temporal virus spread as compared to other production districts, which have a continual and varied host presence.

This paper describes the epidemiology of TYLCV in commercial tomato crops grown in a dry tropical environment over a four-year period soon after its introduction into the region. It also describes the environmental factors influencing whitefly populations, particularly rainfall.

## 2. Results

### 2.1. TYLCV Genetic Diversity

Potential change in virus diversity was monitored by whole genome sequencing of reference TYLCV isolates collected in the Bowen district in 2011, 2013, and 2015. There was very little diversity in the TYLCV genome from these sequenced isolates. Between the 2011 and later isolates, there was 99.6% similarity, and between 2013 and 2015, 99.8%. All three isolates were 98.5% identical to the initial 2006 Brisbane isolate of TYLCV-IL.

### 2.2. TYLCV Monitoring in Crops and Whitefly Populations

The incidence of TYLCV varied spatially and temporally during the study (Table 1 and Table 2, Figure 1) and varied after widespread use of the *Ty-1* resistance gene in 2015. There was a site influence on virus incidence in crop and SLW populations, with sites in area B often having more TYLCV than those in the A or C (Table 1 and Table 2, Figure 1). Sampling time during the season also has an effect, with higher virus incidences generally found late in the season (September/October) compared to early in the season (May) (Table 1 and Table 2). During 2013, TYLCV incidence in the crop was low (0–5.5%) until late in the season, where it was detected at 100% for over half of the sites surveyed. By contrast, in 2014, there was a low virus all season, with TYLCV incidences mostly less than 3% and the highest recorded incidence 28.8%. During 2015, TYLCV was low early but became more widespread towards mid-season, with detections at about half of the sites within the district, at incidences up to 27.5%. By season end, there were moderate to high levels of TYLCV (5.5–100%) at properties in area B, but the virus was not found on sites in the other two areas. TYLCV was detected at two-thirds of the 2016 survey sites, though at low to moderate incidences. The virus was found at less than half of the sites surveyed early-season but at all sites by late season. Early in the 2016 season, half of the survey sites had a susceptible hybrid under cultivation. TYLCV was detected from two of the five sites with a susceptible hybrid and low incidence. Of the remaining five sites, all were Ty-1 hybrids. Interestingly the prevalence and virus incidence were the same, with TYLCV detected at two sites, again at low incidence.

Monitoring vector populations for viruliferous adults revealed variation in relation to the percent of individuals carrying TYLCV. Of note is the variation between the incidence of TYLCV within a tomato crop and the percent of viruliferous SLW collected from that crop (Table 1 and Table 2). Sites where the percent of viruliferous SLW was one-third different from the incidence of TYLCV in the crop was considered evidence of a migratory adult SLW population. There were 31 sites where migratory SLW were detected, and of these, 15 had SLW populations with percent viruliferous SLW greater than in-crop incidence of virus (Table 1, grey shading). These events generally occurred in mid- to late-season and indicated that the primary spread of TYLCV continued through the season, particularly in 2013 and 2014. The incidence of TYLCV in crops also did not correlate with the SLW population level within that crop. There are multiple examples of where the SLW population per leaflet is high, but the TYLCV incidence in the crop is low and vice versa (Table 1). The prevalence of TYLCV within the district decreased by over 17% after the widespread adoption of *Ty-1* hybrids. Before 2015, TYLCV was detected at 35/44 sites (75%) compared to 34/59 sites (57.6%) after 2014. This difference was, however, not statistically significant in a one-way ANOVA test (results not shown).

Interestingly, more migratory populations of SLW with higher percentages of viruliferous than in crop virus incidence occurred in the seasons before 2015 (Table 1—grey shading). Prior to 2014, there was very low use of hybrids with the TYLCV resistance gene *Ty-1*. Large commercial trials of *Ty-1* hybrids occurred in 2014, and the hybrids were almost uniformly adopted in 2015 and 2016, except for a few susceptible crops grown early in 2016. Before 2015, there were 13 examples of migratory SLW with higher viruliferous percentages than in crop virus incidence and only two examples after 2014. Conversely, there were 14 examples of migratory SLW populations with lower viruliferous percentages than virus incidence in crops after 2014, and only two examples before 2015 (Table 1—orange shading). The average percentages of viruliferous SLW across the district were also lower post-2014 (Table 2). The populations with the most viruliferous SLW were the end of season 2013 (64.5%) and mid-late 2014 (16%), whereas, for 2015 and 2016, the highest was 6–10%.

### 2.3. Whitefly Numbers and Rainfall

Evaluation of weather data indicated the amounts and timing of rainfall influences populations of SLW, with wet autumn weather decreasing populations compared to drier conditions. No obvious differences were observed for average minimum or maximum temperatures across the four years of study. During the study, the highest SLW numbers were recorded in late 2015 and early-2016 (Table 2, Figure 2 and Figure 3). The 2013 wet season was relatively normal, with almost 250 mm of rain in February. The continued rain through March and a late rain event of almost 50 mm in May most likely contributed to a low population of SLW in the early production season (Figure 3). The SLW population increased about 3.5-fold by season end. By contrast, 2014 had a relatively dry start, with the monthly average for January almost 150 mm less than in 2013; however, there was a significant rain event (almost 200 mm) in April. This late high rainfall most likely contributed to a low SLW population at the start of the season, as seen in early 2013. The slight recovery of whitefly populations in 2013 but not in 2014 is most likely due to very dry conditions between June and October in 2013. In contrast, for 2014, there were ongoing rain events that continued to suppress the population, with the average population only increasing 2-fold by season end. For 2015, normal rainfall occurred in January, followed by a dry February (average of only *ca* 50 mm) and then almost no further rainfall until the following January. SLW numbers during 2015 were high early, at least 4-fold higher than for the similar time point in 2013 and 2014. The population continued to build at a similar rate to 2013, about 3-fold by season end. The 2015–2016 summer wet season was unusually dry, with less than 50 mm of rainfall until January 2016. The 2016 summer rainfall was lower than previous years, and this, in combination with very high SLW numbers at the end of 2015, contributed to the high whitefly populations detected during May 2016. The high rainfall in June (>100 mm) and following rain in September led to the lower insect populations detected during the remainder of that season. The SLW population decreased by about 4-fold in 2016 by season end compared to 2013 and 2015, which were dry through the season and had increasing SLW populations. No obvious trends at the site within the districts (i.e., A, B, or C) among the population levels of SLW were detected.

## 3. Discussion

The relatively recent introduction of TYLCV into a new production district allowed the spread of a begomovirus to be monitored spatially and temporally in a dry tropics environment which has a clear break in crop production. This provided insight into how quickly the virus established and spread within this environment and how weather influences vector and virus distributions.

The incidence of TYLCV within crops and its prevalence throughout the production district was shown to be influenced by vector populations, crop location, and the use of *Ty-1* virus-resistant hybrids. Vector populations were influenced by the intensity and timing of rainfall and not by location. Dry conditions favoured SLW population increases compared to wet conditions, which do the opposite. There was very limited use of *Ty-1* hybrids pre-2013 or during the 2013 season, and by season end, crops had moderate levels of SLW and high virus incidence. The 2014 season had some *Ty-1* crops and was highly influenced by rainfall which kept SLW numbers at low levels all season, and consequently, TYLCV incidences were also low to moderate. Conversely, the 2015 season and start to the 2016 season had high SLW populations, driven by an unusually dry summer. Most crops in the 2015 and 2016 seasons used *Ty-1* hybrids. The exception was a few crops in early 2016. Although SLW levels were higher during these seasons than observed in 2013, TYLCV incidence was lower, probably due to the widespread adoption of the *Ty-1* hybrids lowering virus spread.

Lapidot et al. [22,23] reported lower TYLCV accumulation and subsequent transmission rates from highly resistant tomato hybrids. The studies also report that adult whiteflies are attracted to virus-infected plants, and once they acquire the virus, they are attracted to uninfected plants. These results were corroborated by Legarrea et al. [24], who reported lower TYLCV accumulation *in planta* in resistant hybrids and changes in SLW feeding preferences based on virus infection. This supports the observed field results from the current study. There was a reduction in the incidence of TYLCV within crops since the widespread adoption of resistant hybrids across the dry tropics district; however, although the incidence was reduced, virus spread continued. The distribution of TYLCV across the district was not overly influenced by using resistant hybrids as the virus was detected at all sites by season end in 2016. There were also fewer detections of migratory viruliferous SLW populations detected after adopting the resistant hybrids. This is partly explained by the lower transmission efficiency from *Ty-1* crops during the season. Once primary spread from weeds has occurred early in the season, the major reservoirs of virus for continued spread during the season are the tomato crops, as multiple sequential plantings are done throughout the season. Where these crops are a resistant hybrid, virus transmission is less efficient, thus reducing the likelihood of detection of migratory viruliferous SLW populations.

Legarrea et al. [24] reported lower virus titre in SLW fed on resistant hybrids compared to susceptible ones, and Srinivasan et al. [25] reported a half to two-thirds lowered acquisition efficiency of TYLCV when SLW fed on *Ty-1* resistant compared to susceptible hybrids. This did not, however, translate to differences in transmission efficiencies in controlled experiments [25]. Srinivasan et al. [25] suggest that under heavy inoculum and vector pressure, it is unlikely that the use of *Ty-1* hybrids will result in a reduction of TYLCV in susceptible hybrids because the resistant ones are a reservoir for the virus and can support SLW populations. The results from the current study indicate this is not the case. It is reasonable to expect that the lowered acquisition efficiencies will impact field transmission due to a high likelihood of SLW populations having a lower proportion of viruliferous adults. Our results show for seasons where *Ty-1* crops were predominantly grown (2015 and 2016), the average percentage of viruliferous SLW was vastly less than detected during seasons of susceptible crops (2013–2014). Our results have also shown lower TYLCV incidences occurred within the district despite high SLW pressure for seasons where *Ty-1* hybrids are mostly grown. Although data on virus incidence in susceptible hybrids is limited, in early 2016, almost half of the survey sites had a susceptible hybrid, and there was either a low incidence of TYLCV or no virus detected from these crops. This was with the highest SLW pressure recorded during the study.

The abundance of alternative hosts for TYLCV during the non-cropping season is also important in understanding virus disease outbreaks. Campbell et al. [21] reported the presence of three important TYLCV weed hosts, *Trianthema portulacastrum*, *Solanum americanum*, and *Amaranthus viridis* in the Bowen production district. Of these, *S. americanum* is a preferred host of SLW in this district, whereas the other two species are not. *S. americanum*, however, is mostly low in abundance during the summer wet season, as is *A. viridis*, whereas *T. portulacastrum* has a high abundance [21]. The major role these weeds have in the epidemiology of TYLCV is through primary movement from the environment into early tomato crops. This type of primary spread is recognized as very important in the development of disease epidemics caused by whitefly-transmitted viruses [26]. There is a clear break in the production of tomatoes over the summer. In seasons where drier conditions coincide with early plantings of tomatoes, the likelihood of migration of SLW from these weed hosts into crops is much higher. Results from vector monitoring activities support this with higher average SLW populations detected early season in 2015 and 2016, compared to 2013 and 2014. Where these weeds also host TYLCV, the flow-on effect is an increased primary spread of the virus. The distribution of weed hosts also partially explains the site differences observed for TYLCV incidence. The virus incidences detected during the study were generally higher for crops grown in area B than elsewhere. The production blocks in this area are surrounded by large tracts of very weedy land, including cattle grazing properties and waterways. The other two areas are less affected by weeds, particularly area C, which is close to a township.

Understanding the impact of a newly introduced begomovirus requires information on virus and vector host ranges, the likely access to host-virus resistance, and the seasonal weather patterns which strongly affect whitefly population dynamics. Although the *Ty-1* hybrids allow profitability to continue in the presence of TYLCV, the introduction of another tomato-infecting begomovirus into Australian production districts would result in major economic losses. It would be unlikely that host resistance would be a viable solution, even if available, as it would be difficult to introgress more genes into a background already requiring resistance to five other pathogens. For other tropical and subtropical Australian vegetable crops, the outcome would be similar due to the low availability of virus resistance genes. Instead, monitoring seasonal weather patterns and strategic deployment of insecticides, biological control agents, and weed control would be essential to allow profitable production to continue.

This study has shown that a begomovirus within a dry tropical environment can easily establish within a single growing season and spread to all areas of susceptible hosts within the district within two growing seasons. It provides very little time for industry to respond to a new disease threat. Further preparedness research is needed to develop strategies to manage begomoviruses. Research is also needed to evaluate other tropical and sub-tropical production districts within Australia to better predict impacts from newly introduced begomoviruses.

## 4. Materials and Methods

### 4.1. Genetic Diversity of TYLCV

Potential change in virus diversity was monitored by whole genome sequencing of reference TYLCV isolates collected in the Bowen district in 2011, 2013, and 2015. Full genome sequences were created by Sanger sequencing overlapping PCRs created using the primer pairs: TY_214_F (5′-TACGAGCCCAATACATT GGGC-3′) and TY_1339_R (5′-GATAAGATTCAACCACAACATCAG-3′), TY1019_F (5′-GCCTCTAATCCAGTGTATGCAACTATG-3′) and TY_2016_R (5′-GAACGTCGT ATCTTCCGCTGCGCGG3′), TY_1759_F (5′-CCGTACTTTGTGTTGCTTTGCCAGTCC-3′) and TY_394_R (5′-GCTGCTGTATGGGCTGTCGAAGTTC-3′).

### 4.2. TYLCV and Vector Monitoring

Surveys of tomato crops were completed from 2013 to 2016. This was done at three-time points, approximately two months apart, during the commercial production period (March to November). Survey sites were spread across the three major production areas within the district, which are approximately 5–20 km from each other (Figure 1).

Survey sites were selected along the edges of crop production blocks, particularly corners where possible, as these are closest to potential environmental reservoirs of TYLCV and SLW. Each survey consisted of a collection of 300 random leaf samples from a designated area delimited by 20 rows and eight trellis panels of tomato, each 3–4 m in length. Except for sites 360 and 376, only 200 samples were collected. The total number of plants present in the designated survey area was approximately 1500 to 2000. Within the same designated area, whitefly was collected from 80–125 sampling points using a vacuum method. Each sampling point was equivalent to an area of four tomato leaflets (equivalent to one compound leaf). Crops were only monitored for TYLCV as there are no other SLW-vectored viruses affecting tomatoes in this district.

### 4.3. Sample Indexing

Total nucleic acid extracts (TNAEs) were prepared from tomato leaf samples using the Qiagen Biosprint Kit as per manufacturers’ instructions and from whitefly samples using the method by [27]. Both plants and whiteflies were tested for TYLCV using a specific PCR assay [7]. Leaf samples were tested as bulks of 10 individual samples per survey site. Whitefly samples were also collected tested individually (up to 100) per site, or as 20 bulks of 5. For sites 545 and 555, where all bulks of 20 tested positive, a further 25 individuals were tested separately to better estimate the percent of viruliferous adults in the population. In 2016, additional SLW adults were tested at some sites where TYLCV wasn’t detected in the first 100 adults tested, and virus incidence within the crop was at least 5%. For these sites, an additional 100 adults were tested in 20 bulks of 5. For samples where individuals were bulked together for molecular indexing, the percent incidence is given as a statistical estimate derived using a statistical model developed by M. Sharman (pers. Comm.), based on [28,29]. The whitefly present in the district were morphologically confirmed as SLW from representative samples collected during the study.

### 4.4. Weather Monitoring

Weather data from the nearby Australian Bureau of Meteorology stations was sourced for the study period. The locations are shown in Appendix A. Maximum and minimum temperature and rainfall were analysed for the wet and dry seasons. The levels of whitefly were monitored through the season and mapped against weather data.

## Figures and Tables

**Figure 1 plants-11-00776-f001:**
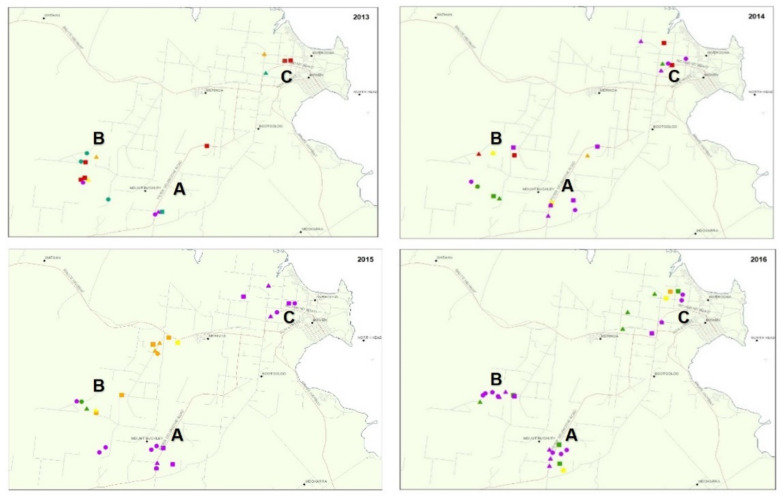
Distribution and severity of tomato yellow leaf curl virus (TYLCV) in the study area from 2013–2016. The three distinct survey areas are denoted as A, B, and C. Survey time points during the season are depicted as early = dot, mid = triangle, and late = square and TYLCV incidence in crops (shown as a percentage in brackets) by colour with no virus = purple, low (0.1–5%) = green, mod (5.1–10%) = yellow, high (10.1–30%) = orange and very high (>30%) = red.

**Figure 2 plants-11-00776-f002:**
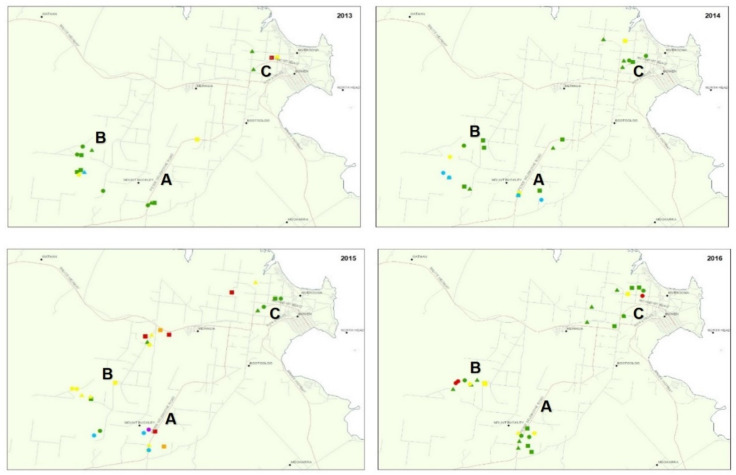
Distribution and population density of silverleaf whitefly (SLW) in the study area for years 2013 -2016. The three distinct survey areas are denoted as A, B, and C. Survey time points during the season are depicted as early = dot, mid = triangle, and late = square and population densities by number of adult SLW per leaflet indicated in brackets and colour coded with no SLW = purple, very low (0.01–0.09) = light blue, low (0.1–1.0) = green, mod (1.1–3.0) = yellow, high (3.1–5.0) = orange, very high (>5.0) = red.

**Figure 3 plants-11-00776-f003:**
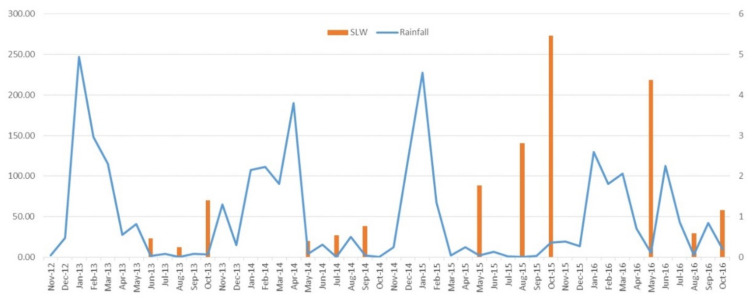
Graph of rainfall and silverleaf whitefly (SLW) populations during the study period. Rainfall is shown in mm on the left *y*-axis and SLW populations as adults per leaf on the right *y*-axis. Data is provided as a monthly average.

**Table 1 plants-11-00776-t001:** List of sites surveyed for silverleaf whitefly (SLW) and tomato yellow leaf curl virus (TYLCV) since June 2013. The incidence of TYLCV in crop and adult SLW was assessed through molecular indexing is provided for each site, in addition to the number of SLW collected and the estimated population size of SLW per leaflet. Sites were identified to have migratory SLW populations if there was a poor correlation between TYLCV detections in SLW and the crop. These are indicated by grey shading where the TYCV-SLW is >one-third of the TYLCV -crop incidence and orange shading where the reverse occurs. For sites where the TYLCV incidence is <5% or the proportion of TYLCV-SLW are considered to correlate well with TYLCV-crop incidences (i.e., values < one-third different), there is no shading, and data was not evaluated for correlations. Instances where data was not determined are indicated by nd.

Survey Month and Year	Survey Area	Survey Site	Varietal Susceptibility (S) or Resistance (R) ^1^ to TYLCV	TYLCV Incidence ^2^		SLW Population Size per Leaflet ^4^
				Crop (%) ^3a^	SLW (%) ^3a^	
Jun-13	A	272	nd	0.3	0.0	0.2
	B	266	nd	1.4	1.0	0.4
		267	nd	2.6	0.0	1.1
		268	nd	0.3	0.6	0.3
		273	S	1.4	1.5	0.3
Aug-13	A	283	nd	0.0	0.0	0.2
	B	277	R	2.6	6.0	0.4
		278	R	1.4	57.5	0.1
		280	R	5.5	13.0	0.3
	C	275	S	0.3	1.0	0.2
		284	nd	2.2	20.0	0.2
Oct-13	A	297	S	100.0	40.0	1.5
		298	R	3.5	4.0	0.6
	B	294	S	100.0	99.0	0.4
		295	R	100.0	49.0	0.6
		296	nd	100.0	95.0	0.2
	C	290	R	12.4	67.0	1.3
		291	nd	20.6	97.8	5.4
May-14	A	300	R	0.4 (0.01–1.9)	0.0	0.1
		301	R	0.0	0.0	0.1
	B	302	S	2.2 (0.8–4.7)	2.2	0.1
		303	S	1.4 (0.4–3.6)	0.0	0.1
		305	S	11.3 (6.8–17.4)	32.0	1.2
		306	S	0.4 (0.01–1.9)	8.0	0.9
	C	307	nd	0.0	0.0	0.4
		308	nd	0.7 (0.1–2.5)	0.0	0.3
July-14	A	359	R	0.0	0.0	0.01
		360	R	0.4 (0.01–1.9) ^3b^	35.0	0.7
		361	S	0.0	16.0	0.6
	B	355	R	0.0	1.3	0.1
		356	S	14.9 (9.1–22.6)	28.0	0.4
		357	R	28.8 (16.1–51.0)	38.3	0.1
		358	S	0.4 (0.01–1.9)	6.0	2.0
	C	352	S	0.0	0.0	0.2
		353	S	0.4 (0.01–1.9)	0.0	0.1
		354	S	2.2 (0.8–4.7)	2.7	0.1
Sept-14	A	382	R	0.0	6.0	3.0
		383	R	0.0	0.0	0.4
		384	R	0.4 (0.1–1.9)	0.0	0.1
	B	378	R	0.0	3.1	0.2
		381	R	6.1 (3.3–10.1)	31.0	0.6
		385	R	23.7 (14.0–38.1)	0.0	0.5
	C	376	R	0.5 (0.01–2.8) ^3b^	38.0	1.7
		377	R	0.0	34.0	0.5
May-15	A	542	R	0.0	0.0	0.01
		543	R	0.0	0.0	0.0
		544	R	0.0	0.0	0.01
	B	537	R	0.0	0.0	0.3
		538	R	0.0	20.0	0.01
		539	S	0.0	0.0	3.0
		540	S	3.0 (1.3–6.0)	8.3 (3.3–16.0) ^3c^	1.5
		541	R	0.0	0.0 ^3c^	2.0
		545	S	12.4 (7.5–18.9)	16.0 ^3d^	2.0
	C	535	S	0.0	1.5	0.1
		536	R	0.0	0.0 ^3c^	0.5
Aug-15	A	552	R	0.4 (0.01–1.9)	1.0 (0.3–5.6) ^3c^	3.0
		553	R	0.0	0.0 ^3c^	1.5
	B	550	R	4.0 (1.9–7.2)	0.0 ^3c^	3.0
		551	R	8.8 (5.1–13.8)	2.1 (0.3–7.3) ^3c^	2.0
		554	S	27.5 (14.7–49.9)	2.1 (0.3–7.3) ^3c^	2.0
		555	S	13.5 (8.3–20.6)	36.0 ^3d^	0.7
	C	556	R	0.0	1.0 (0.3–5.6) ^3c^	1.5
		557	R	0.0	0.0	0.1
Oct-15	A	564	R	0.0	0.0	0.2
		565	R	0.0	1.0 (0.3–5.6) ^3c^	2.3
	B	558	R	28.8 (16.1–51.0)	5.6 (1.8–12.6) ^3c^	9.4
		559	R	13.5 (8.3–20.6)	0.0 ^3c^	9.4
		560	R	14.9 (9.1–22.6)	0.0 ^3c^	4.7
		561	R	100 (19.4–100.0)	8.3 (3.3–16.0) ^3c^	7.8
		562	R	10.4 (6.2–16.1)	2.1 (0.3–7.3)	4.7
		563	R	5.5 (2.9–9.4)	2.1 (0.3–7.3) ^3c^	7.8
	C	566	R	0.0	2.1 (0.3–7.3) ^3c^	7.8
		567	R	0.0	1.0 (0.3–5.6) ^3c^	0.5
May-16	A	577	R	0.0	0.0	0.5
		578	R	0.0	0.0	1.5
		580	R	0.3 (0.01–1.9)	0.0	0.8
	B	572	R	0.3 (0.01–1.9)	0.0	1.9
		573	R	0.0	0.0	0.2
		574	S	6.1 (3.3–10.1)	0.0	12.1
		576	S	0.7 (0.1–2.5)	0.0	11.5
	C	571	S	0.0	0.0	11.5
		579	S	0.0	0.0	0.1
		582	S	0.0	0.0	1.0
Aug-16	A	583	R	1.05 (0.2–3.0)	0.0	3.0
		584	R	0.0	0.0	0.4
		585	R	3.05 (1.3–6.0)	0.0	0.5
	B	586	R	0.0	00.0	0.2
		587	R	0.0	1.02 (0.03–5.5)	0.6
		588	R	1.8 (0.6–4.2)	0.0	0.7
	C	589	R	7.3 (4.1–11.8)	0.0 ^5^	0.5
		590	R	9.5 (5.6–14.9)	0.0 ^5^	0.9
		591	S	14.9 (9.1–22.6)	0.0 ^5^	0.9
		592	nd	18.2 (11.1–27.9)	0.0 ^5^	0.4
Oct-16	A	599	R	5.0 (2.5–8.6)	3.2 (0.6–9.1)	0.8
		600	R	5.5 (2.9–9.4)	8.2 (3.3–16.4)	0.7
		601	R	2.6 (1.0–5.3)	4.4 (1.2–10.8)	0.4
	B	606	R	23.7 (14.0–38.1)	0.0 ^5^	1.5
		607	R	23.7 (14.0–38.1)	0.0 ^5^	1.8
		608	R	0.34 (0.01–1.87)	0.0	3.0
	C	602	R	7.34 (4.12–11.84)	5.6 (1.8–12.6)	1.4
		603	R	0.34 (0.01–1.87)	11.3 (5.1–26.6)	0.6
		604	R	1.42 (0.38–3.61)	1.0 (0.03–5.5)	0.4
		605	R	28.8 (16.1–51.0)	0.0	0.2

^1^ R—indicates the presence of the *Ty-1* gene in the hybrid. ^2^ For samples where individuals were bulked together for molecular indexing, the percent incidence is given as a statistical estimate, and the range is given in parentheses. ^3a^ For all sites, 300 leaf samples were tested as 30 bulks of 10 individual samples, and whiteflies were tested individually (up to 100) unless otherwise indicated. ^3b^ Plant samples were tested as 20 bulks of 10 at this site, and ^3c^ whitefly samples bulked at 5 and 20 bulks tested per sample. ^3d^ For whitefly samples where all bulks were positive, a subsample of 25 individuals was tested separately. ^4^ The whitefly population size per tomato leaflet was calculated by dividing the number of whitefly individual adults collected by the number of sampling sites divided by 4, as each vacuum site is estimated to cover approximately four tomato leaflets. ^5^ Sites where an additional 100 WF were indexed for TYLCV, in total 200 WF were tested from each of these sites, bulked by 5.

**Table 2 plants-11-00776-t002:** Summary of tomato yellow leaf curl virus (TYLCV) incidence in crops and silverleaf whitefly (SLW) populations and the number of adult SLW detected during surveys. The total number of sites surveyed in each area (A, B, and C) and for each time point is listed with the prevalence of TYLCV, the average percent of viruliferous SLW, and average SLW per leaflet listed across the sites. The incidence of TYLCV in crops and SLW populations is data given as the range from minimum to maximum. Data where TYLCV incidence is 20% or greater is highlighted in bold.

Year	Season	Total Survey Sites	Prevalence of TYLCV ^1^	TYLCV Incidence in Crops (%) at Each Study Location within the District			Average Viruliferous SLW (%)	TYLCV Incidence in SLW (%) across the District			Average Number of SLW per Leaflet ^2^	Number of SLW per Leaflet ^3^		
				A	B	C		A	B	C		A	B	C
2013	Early-June	5	5	0.34	0.3–2.6	Nd ^1^	0.6	0.0	0.0–1.5	nd ^1^	0.5	0.2	0.3–1.1	nd ^1^
	Mid-August	6	5	0.0	1.4–5.5	0.3–2.2	16.2	0.0	**6.0–57.5**	1.0–20.0	0.2	0.2	0.1–0.4	0.2–0.2
	Late-October	7	7	**3.5–100.0**	**100.0**	**12.4–20.6**	64.5	**4.0–40.0**	**49.0–99.0**	**67.0–97.8**	1.4	0.6–1.5	0.2–0.6	1.3–5.4
2014	Early-May	8	6	0.0–0.4	0.4–11.3	0.0–0.7	5.1	0.0	**0.0–32.0**	0.0	0.4	0.1	0.1–1.2	0.3–0.4
	Mid-July	10	6	0.0–0.4	**0.0–28.8**	0.0–2.2	15.9	**0.0–35.0**	**1.3–38.3**	0.0–2.7	0.5	0.01–0.7	0.1–2.0	0.1–0.2
	Late-September	8	4	0.0–0.4	0.0–23.7	0.0–0.5	16.0	0.0–6.0	**0.0–31.0**	**34.0–38.0**	0.8	0.1–3.0	0.2–0.6	0.5–1.7
2015	Early-May	11	2	0.0	0.0–12.4	0.0	3.7	0.0	**0.0–20.0**	0.0–1.5	1.8	0.0–0.01	0.01–3.0	0.1–0.5
	Mid-August	8	5	0.0–0.4	**4.0–27.5**	0.0	6.0	0.0–1.0	**0.0–36.0**	0.0–1.0	2.8	1.5–3.0	0.7–3.0	0.1–1.5
	Late-October	10	6	0.0	**5.5–100.0**	0.0	2.2	0.0–1.0	0.0–8.3	1.0–2.1	5.4	0.2–2.3	4.7–9.4	0.5–7.8
2016	Early-May	10	4	0.0–0.3	0.0–6.1	0.0	0.0	0.0	0.0	0.0	4.37	0.8–1.5	0.2–12.1	0.1–11.5
	Mid-August	10	7	0.0–3.0	0.0–1.8	7.3–18.2	0.1	0.0	0.0–1.0	0.0	0.59	0.4–3.0	0.2–0.7	0.4–0.9
	Late-October	10	10	2.6–5.5	**0.3–23.7**	**0.3–28.8**	9.9	3.2–8.2	0.0	0.0–11.3	1.16	0.4–0.8	1.5–3.0	0.2–1.4

^1^ Prevalence of TYLCV across the district per survey time point (number of positive sites); ^2^ Average viruliferous SLW across the district per survey time point (%); ^3^ Average number of SLW per leaflet across the district per survey time point.

## Data Availability

Data is contained within the article or Appendix A.

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
