# Peer review of "The Establishment and Spread of a Newly Introduced Begomovirus in a Dry Tropical Environment Using Tomato Yellow Leaf Curl Virus as a Case Study"

_plants, 2022, doi:10.3390/plants11060776_

Round 1

Reviewer 1 Report

Please see the annotated pdf uploaded.

Author Response

Authors are advised to mention the host (main if many) host(s) affected by these viruses

Have edited the text to say none infect tomato. Listing whole host ranges isn’t relevant.

Since Ty-1 resistance is mentioned in the text it should be briefly introduced.

Amended text.

Authors need to remind briefly the reader when Ty-1 hybrids were introduced in these areas. The last sentence of the paragraph comes to small surprise.

Amended text.

Lengthy title. Authors are advised to shorten

Disagree as it’s a complicated table that requires explanation.

What do grey and orange highlighted cells designate? What do bold lettered numbers designate?

They are mentioned in the caption but the blue changed to grey in this version. The caption is edited to grey. The bold is also mentioned in the caption.

this needs clarification in the writingline 161.

Assume this refers to the use of in-crop incidence but not sure. The other 3 reviewers were happy with the way it is currently written.

too small to distiguish the difference – line 217

This was addressed by enlarging the figures in the manuscript.

Minor typos through document – amended where needed, some suggestions were not correct

What do authors refer to with this expression? – line 229

This is normal convention when talking about horticulture production cycles

this needs clarification in the writing – line 247

It refers to the refers in the preceding sentence as per normal writing conventions.

Reviewer 2 Report

The MS presents a comprehensive and interesting study on the spreading of a newly introduced begomovirus in a dry tropical environment. The overall work is good and worthy to publish in present form.

Author Response

We thank the reviewers for their positive comments. 

Reviewer 3 Report

In this manuscript Gambley and co-authors described the epidemiology of the geminivirus Tomato yellow leaf curl virus (TYLCV) in commercial tomato crops over four-year period in three distinct survey areas from Australia. In this study the authors provide valuable information about TYLCV-Israel both spatial-temporal incidence. Also, they recorded the population dynamics of the TYLCV insect vector Bemisia argentifolii (silverleaf whitefly, SLW). Their data demonstrated the widespread adoption of the Ty-1 hybrids reducing virus spread. The authors found that the environmental factors influencing SLW populations, in special rainfall.

I think that this manuscript is suitable for the journal scope however I have some minor comments and suggestions:

  • Authors could comment if they detected or monitor other viruses spread by SLW in the sampled tomato plants.
  • Authors observed tomato plants with the second most prevalent geminivirus Tomato leaf curl virus (ToLCV)?
  • Do the authors recorded the severity of symptoms? If yes, the authors can mentioned it in the text.
  • Table 1 can move to supplemental data and leave the table 2 in the main text.
  • Improve the quality of image quality, enlarge a little bit the shapes and names districts in the map.

Author Response

  • Authors could comment if they detected or monitor other viruses spread by SLW in the sampled tomato plants.

Australia has a very low diversity of begomoviruses and there are no other SLW viruses that affect tomato in that district. Details added to the methods section to capture this.

  • Authors observed tomato plants with the second most prevalent geminivirus Tomato leaf curl virus (ToLCV)?

ToLCV is restricted to northern Australia and not found in commercial crops. Have added this detail to the introduction.

  • Do the authors recorded the severity of symptoms? If yes, the authors can mentioned it in the text.

No symptom severity wasn’t recorded.

  • Table 1 can move to supplemental data and leave the table 2 in the main text.

This table gets referred to multiple times in the text and as such should remain in the main text. The other two reviewers were ok with it being in the main text.

  • Improve the quality of image quality, enlarge a little bit the shapes and names districts in the map.

Agree and have enlarged the images in the manuscript

Reviewer 4 Report

1:Why this old data is being presented (2013-2016), its 6 years gap?

2: Authors should perform the latest sampling ad data recording because in this 6 years many environmental factors changed and there might be a change in the SLW spread and TYLCV incidence in the sampling areas and there may be more TYLCV incidence.

3: Authors should mention the months of sampling in the year 2013.

4: Authors should mention the incidence of SLW in 2013.

5: Did the authors detected the TLYCV in SLW from sampling areas? if yes then what was the percentage of virus in those SLW.

6: Authors have mentioned a low incidence of TYLCV in susceptible hybrids? How it is possible?

7: Changes in the weather conditions and temperature affects the incidence of SLW spread and TYLCV on the crops then authors should discuss in the text.

8: Is there any collection of samples from weeds growing near sampling areas? is there any detection of virus from these samples?

Author Response

1:Why this old data is being presented (2013-2016), its 6 years gap?

We didn’t get an opportunity to publish any earlier due to work commitments in other research projects.

2: Authors should perform the latest sampling ad data recording because in this 6 years many environmental factors changed and there might be a change in the SLW spread and TYLCV incidence in the sampling areas and there may be more TYLCV incidence.

Our research is linked to available funding and there was no further funding for this work post 2016.

3: Authors should mention the months of sampling in the year 2013.

This is listed in Table 1.

4: Authors should mention the incidence of SLW in 2013.

This is listed in Table 1.

5: Did the authors detected the TLYCV in SLW from sampling areas? if yes then what was the percentage of virus in those SLW.

This is listed in Table 1.

6: Authors have mentioned a low incidence of TYLCV in susceptible hybrids? How it is possible?

There are multiple reasons why a crop may have a low virus incidence, even if it is susceptible. In this study low incidence in susceptible crops in 2016 was most likely linked to widespread use of the Ty-1 genotype reducing overall inoculum in the district. This is discussed in the paper.

7: Changes in the weather conditions and temperature affects the incidence of SLW spread and TYLCV on the crops then authors should discuss in the text.

Temperature was very stable during the study thus not analysed further. The influence of rainfall is discussed.

8: Is there any collection of samples from weeds growing near sampling areas? is there any detection of virus from these samples?

The weed host range is mentioned in the paper in the discussion section. This data was previously published and not relevant to this study.